# Oxidative Stress Management in Chronic Liver Diseases and Hepatocellular Carcinoma

**DOI:** 10.3390/nu12061576

**Published:** 2020-05-28

**Authors:** Daisuke Uchida, Akinobu Takaki, Atsushi Oyama, Takuya Adachi, Nozomu Wada, Hideki Onishi, Hiroyuki Okada

**Affiliations:** Department of Gastroenterology and Hepatology, Okayama University Graduate School of Medicine, Dentistry and Pharmaceutical Sciences, Okayama 700-8558, Japan; d.uchida0309@gmail.com (D.U.); at841205@gmail.com (A.O.); adataku719@yahoo.co.jp (T.A.); nonsan0808@yahoo.co.jp (N.W.); ohnis-h1@cc.okayama-u.ac.jp (H.O.); hiro@md.okayama-u.ac.jp (H.O.)

**Keywords:** oxidative stress, chronic hepatitis, hepatocellular carcinoma

## Abstract

Chronic viral hepatitis B and C and non-alcoholic fatty liver disease (NAFLD) have been widely acknowledged to be the leading causes of liver cirrhosis and hepatocellular carcinoma. As anti-viral treatment progresses, the impact of NAFLD is increasing. NAFLD can coexist with chronic viral hepatitis and exacerbate its progression. Oxidative stress has been recognized as a chronic liver disease progression-related and cancer-initiating stress response. However, there are still many unresolved issues concerning oxidative stress, such as the correlation between the natural history of the disease and promising treatment protocols. Recent findings indicate that oxidative stress is also an anti-cancer response that is necessary to kill cancer cells. Oxidative stress might therefore be a cancer-initiating response that should be down regulated in the pre-cancerous stage in patients with risk factors for cancer, while it is an anti-cancer cell response that should not be down regulated in the post-cancerous stage, especially in patients using anti-cancer agents. Antioxidant nutrients should be administered carefully according to the patients’ disease status. In this review, we will highlight these paradoxical effects of oxidative stress in chronic liver diseases, pre- and post-carcinogenesis.

## 1. Introduction

With the progression of viral hepatitis treatment, hepatitis C virus (HCV) can now be eradicated in >90% of treated patients, and hepatitis B can be controlled with low serum viral DNA, although viral eradication is still difficult [1,2]. The number of newly diagnosed hepatocellular carcinoma cases is decreasing; however, there are still many new patients, due to chronic viral hepatitis B and C [3]. The recent trend in increased obesity indicates that chronic viral hepatitis-related carcinogenesis is also partly due to non-alcoholic fatty liver disease (NAFLD) coexistence.

NAFLD can progress to HCC as a single disease or be concomitant with chronic viral hepatitis. It is mainly dependent on lifestyle changes, such as the spread of a Westernized diet and transportation progression from walking to a motorized society. As the numbers of obese people increase, so too does NAFLD, in addition to diabetes. Although Asian people are usually not very obese, they can develop obesity-related disease, such as diabetes, without severe obesity.

Present issues with the management of NAFLD can be summarized as follows: the natural history of the disease is not well defined, high-risk patients cannot be easily identified, and promising treatment protocols for the different stages of the disease are unclear. There have been many studies related to these problems; however, the natural history and the optimal management strategy at different stages, from simple steatosis, steatohepatitis, liver cirrhosis, HCC and beyond, are not clear [4,5,6]. The definition of NAFLD is under reconsideration, because patients with a history of alcohol consumption and those who habitually consume low amounts of alcohol are not definite non-alcoholic patients. Recently, a new disease category, metabolic-dysfunction-associated fatty liver disease (MAFLD) is advocated for patients with hepatic steatosis, in combination with one of the following three criteria: overweight/obesity, presence of type 2 diabetes mellitus, evidence of metabolic dysregulation [7]. Although the definition of MAFLD excludes individuals with excessive alcohol consumption, current drinkers and patients with a history of drinking can be included. The characteristics of the patients, natural history, and treatment strategy will change according to the new criteria. Given that the natural history might continue for several decades, a large cohort study should be conducted of individuals with NAFLD or MAFLD, and in vivo models should be used to predict the clinical course over decades.

Viral hepatitis-related inflammation and the fat-related stress response are both correlated with oxidative stress, a toxic response related to disease progression. However, oxidative stress also functions as an “anti-cancer” response that should be maintained in order to combat cancer [8]. In this review, we discuss the role of oxidative stress in chronic viral hepatitis and NAFLD, before and after carcinogenesis.

## 2. Oxidative Stress in the Pathogenesis of Chronic Liver Diseases and Hepatocarcinogenesis

### 2.1. Oxidative Stress in the Cellular Pathogenesis of Chronic Liver Diseases

Reactive oxygen species (ROS) are a source of oxidative stress generated in various organelles and stress pathways, such as mitochondria, peroxisomes, and the endoplasmic reticulum (ER) [9]. Mitochondria produce cellular ROS due to inefficiencies in electron flow along the electron transport chain (ETC). Under physiological conditions, the majority of incompletely reduced ROS, such as superoxide, are detoxified into water, and steady state necessary oxidant concentrations are maintained at relatively low levels (<1% of total oxygen consumed by mitochondria), via various antioxidant defenses and repair enzymes [10]. Physiological levels of ROS are necessary as a self-guarding response against damage from a toxic microbiome and as a plasma membrane repair response [11]. Excessive superoxide levels are produced within injured mitochondria through electron leakage, and then converted to hydrogen peroxide (H_2_O_2_) by superoxide dismutase (SOD). There are several antioxidant enzymes, including glutathione peroxidase (GPx) or catalase that can metabolize H_2_O_2_ to non-toxic H_2_O. There are also toxic hydroxyl radical producing reactions, namely the Fenton and/or Haber–Weiss reactions. Peroxisomes play a major role in fatty acid metabolism and produce oxidative stress via similar enzymatic pathways to mitochondria [12]. ER produces ER-stress associated oxidative stress when it contains the enzymes endoplasmic reticulum oxidoreductin 1 α and protein disulfide isomerase [13]. These oxidants produce many toxic factors, such as TonEBP, resulting in cell injury and inflammation [14]. As all of these organelles produce ROS, there are many approaches to antioxidant therapy targeting these pathways, including the administration of inhibitors of mitochondrial dysfunction, ER stress, and NADPH oxidases (NOX) inhibitors; these approaches show varying degrees of clinical effectiveness [9].

In hepatocytes, ROS are produced by free fatty acid metabolism in mitochondria and microsomes [15]. Hepatocyte apoptosis due to inflammation also produces ROS, resulting in the inflammatory and fibrotic responses of Kupffer cells and hepatic stellate cells (HSCs).

Hepatic non-parenchymal cells, such as hepatic stellate cells (HSCs), liver resident macrophages (Kupffer cells), fibroblasts, and liver sinusoidal endothelial cells (LSECs), have been shown to be correlated with chronic inflammation and oxidative stress [16,17].

In liver fibrosis progression, the activation of HSCs into contractile and matrix-producing myofibroblasts is a central role. HSCs can be activated by mediators, such as transforming growth factor β1 (TGF-ββ1) secreted from damaged hepatocytes, activated Kupffer cells, and aggregated platelets. TGF-β1 plays key roles in regulating myofibroblasts, which induce continuous extracellular matrix (ECM) deposition, resulting in liver fibrosis [18]. Recently, cytoglobin has been shown to be expressed in HSCs, which catabolize H_2_O_2_ and lipid hydroperoxides, molecules involved in HSC activation [19]. Cytoglobin acts as an antioxidant in HSCs, and its knockout mouse model exhibited the progression of NASH and an extremely high incidence of HCC [20]. In addition, the selective overexpression of cytoglobin in HSCs has been shown to attenuate liver fibrosis in a mouse model [21]. TGF-β1 suppresses human cytoglobin expression through phosphorylated SMAD2 and the M1 repressor isoform of SP3, suggesting its key role in oxidative stress control in HSCs [22].

Oxidative stress control in HSCs has been demonstrated via several approaches. In multidrug resistance gene 2 knockout (Mdr2-/-) mice, a genetic model that resembles primary sclerosing cholangitis (PSC), portal fibroblasts and HSCs were both activated; while they became inactive, resulting in the improvement of disease, with the administration of antioxidant NOX inhibitor [16]. HSCs are stimulated by ferric chloride and citrate addition followed and revert to a quiescent status with iron chelators, suggesting that iron-related oxidative stress is involved in HSC activation [23].

Kupffer cells that are activated by inflammatory cytokines, such as interleukin 1β (IL1β) released from damaged hepatocytes, subsequently activate NFκB, resulting in the release of IL6 and hepatocyte transforming oncogenic pathways, such as STAT3 [24]. One of the NFκB regulating oncogenes, Astrocyte elevated gene-1 (AEG-1), has been shown to be involved in hepatocarcinogenesis and the knock-out of hepatocyte AEG-1 resulted in the improvement of hepatocarcinogenesis in vivo and increased sensitivity to oxidative stress in vitro, while the knockout of macrophage AEG-1 resulted in the improvement of hepatocarcinogenesis and functional anergy in vitro, indicating that such genes cause both hepatocytes and Kupffer cells to cause hepatocarcinogenesis [25].

LSECs form hepatic sinusoids and control sinusoidal blood flow and the activation of the neighboring HSCs. LSECs are recognized as gatekeepers of hepatic inflammatory reactions, guarding Kupffer cells and HSCs from aberrant activation via toxic molecules in the portal vein [26]. In chronic liver diseases, inflammation gives LSECs physiological shear stress and capillarization, resulting in the activation of HSCs to cause liver fibrosis [17].

### 2.2. Overview of the Oxidative Stress in the Pathogenesis of Hepatocarcinogenesis

Chronic strong exposure to ROS induces chronic inflammatory disease progression and carcinogenesis [27,28]. ROS has been shown to be correlated with digestive system cancers, such as gastrointestinal cancer, cholangiocarcinoma, pancreatic cancer and HCC [29,30,31]. HCC develops from chronic viral hepatitis and non-alcoholic steatohepatitis (NASH), which is a severe form of NAFLD [15,32]. The mechanisms underlying hepatocarcinogenesis have been thoroughly investigated in chronic viral hepatitis-related HCC, as chronic viral hepatitis has been recognized as a major cause of HCC for several decades. The representative mechanisms underlying HBV- and HCV-related chronic liver disease progression and hepatocarcinogenesis have been shown to involve viral protein’s functions, such as immune interference, tumor initiation or tumor suppression interference and oxidative stress response induction [33]. As chronic viral hepatitis-related liver inflammation is controllable with anti-viral agents, residual liver fibrosis-related cytokines or stress response and steatosis-related oxidative stress have a strong influence on the present and future incidence of hepatocarcinogenesis.

### 2.3. Chronic Viral Hepatitis and Hepatocarcinogenesis

Recent advances in anti-viral treatment have enabled us to eradicate HCV-RNA and reduce HBV-DNA in serum to the lower limit of normal levels. Although HBV viral DNA can be reduced to undetectable levels in the blood with nucleos(t)ide analogues, HBV virion and translated viral proteins are able to persist in infected hepatocytes and surrounding immune cells, even under a clinical cure, with seroclearance of the HBV envelope antigen (HBsAg) and the emergence of anti-HBs antibodies [34]. Although direct-acting antivirals (DAA) can induce a sustained virologic response (SVR) in >95% of chronic hepatitis C patients, the risk of HCC—despite being reduced—still exists [35].

#### 2.3.1. HBV-Related Chronic Hepatitis

##### Recent Advances in the Management of HBV-Related Chronic Hepatitis

HBV has been well-controlled with the administration of nucleos(t)ide analogues for nearly two decades. However, HCC can occur in patients with well-controlled viremia. There are many reports on predicting the risk of HCC development in patients treated with nucleos(t)ide analogues. The PAGE-B score, which includes baseline age, sex, and platelet count, could discriminate against Caucasian patients with 5-year HCC incidence rates of 0%, 3%, and 17% [36]. In Asian patients, the modified PAGE-B, which includes the serum albumin level in addition to the PAGE-B variables, showed better predictive performance than PAGE-B and four other scores: REACH-B (variables: male sex, age, alanine aminotransferase, HBeAg status, and HBV-DNA), GAG-HCC (variables: age, male sex, HBV-DNA, HBV core promoter mutations, cirrhosis), CU-HCC (variables: age, albumin, bilirubin, HBV-DNA, cirrhosis), and PAGE-B [37,38]. Basically, the patients who develop HCC even when hepatitis was controlled with nucleos(t)ide analogues were elderly male patients with advanced liver fibrosis and liver reservoir dysfunction. The screening of patients according to these predictive scores is recommended to facilitate the strict follow-up of these high-risk patients and save the costs associated with unnecessary imaging. In addition, Asian patients with chronic hepatitis B easily develop metabolic syndrome and a steatosis-related transaminase increase, even with a low BMI, indicating an additional HCC risk increase [39].

Given that one of the strong predictive markers for HCC is a low platelet count, which reflects liver fibrosis in cirrhosis, other markers of liver fibrosis have been investigated. Liver stiffness has recently been acknowledged to be a reliable non-invasive marker (measured by ultrasound), that can be used to diagnose liver fibrosis. Given that an accurate assessment of liver fibrosis is essential for surveillance and management, the evaluation of liver stiffness is strongly recommended in many guidelines [40,41]. Transient elastography (TE) and advanced two-dimensional (2D) shear wave elastography (SWE) are now widely used to assess liver stiffness. Higher liver stiffness has been shown to predict HCC in nucleos(t)ide analogue-treated patients, in addition to cirrhosis, old age, male sex, lower platelet count and lower albumin level [42]. A new deep learning radiomics elastography system that adopts a radiomic strategy for the quantitative analysis of heterogeneity in 2D-SWE, has shown a much better diagnostic accuracy than 2D-SWE in chronic hepatitis B [43]. With these new non-invasive diagnostic tools, we can narrow down the HBV-positive patients with a high risk of developing HCC, and thereby reduce costs and improve quality of life.

##### Oxidative Stress in the Pathogenesis of HBV-Related Chronic Hepatitis and HCC

The mechanisms underlying hepatocarcinogenesis include the direct effect of HBV-producing proteins and integration of the HBV genome into human genome *cis-* or *trans-* activating targeted oncogenic or anti-oncogenic genes. There are many genes that have been reported to be affected by HBV. A genome sequencing analysis revealed that most (84%) HBV-infected HCC patients exhibited HBV DNA integration into the host genome, leading to cis activation or inactivation of cancer regulatory genes, and resulting in HCC development [44].

One of the featured pathways of HBV-HCC pathogenesis involves direct tumorigenesis-related genes, such as tumor suppressor gene *TP53* and *CDKN2A* [45]. CDKN2A encodes cyclin-dependent kinase (CDK) inhibitor 2A, while the mammalian cell cycle controlling enzyme CDK2 has been shown to be correlated with HCC. CDK2 is regulated by E-type cyclins E1 and E2. The *CCNE1* locus, which encodes cyclin E1, has also been identified as an HBV integration site in 2–5% of HCC patients. Loss of Cyclin E1 attenuated hepatitis and HCC development in a mouse model [46]. Such oncogenic gene regulation change is believed to be a leading mechanism underlying HBV-related hepatocarcinogenesis, especially in non-cirrhotic HBV-related HCC [47].

Of the significantly mutated exome sequence in HBV-related HCC patients, the oxidative stress- related KEAP1 has been shown to be involved, although the percentage was not very high (12%, versus 61% for telomere maintenance-related genes, 54% for Wnt-**β**-catenin signals and 51% for PI3K-AKT-mTOR pathway signals) [45,48]. The KEAP1 gene encodes Keap1 protein, which binds to the antioxidant-inducing transcriptional factor nuclear factor erythroid 2-related factor (Nrf2). Under oxidative stress conditions, Nrf2 is released from Keap1 and translocates to the nucleus, resulting in antioxidant defense-related protein activation. This pathway activation has been reported to be a mechanism through which cancer cells escape from toxic oxidative stress or other stress responses, such as endoplasmic reticulum (ER) stress. One of the Keap1 regulatory protein tripartite motif-containing (TRIM)25 has been shown to be upregulated by ER stress, resulting in Nrf2 release followed by escape from cytotoxic ER stress and the survival of tumor cells [49]. A high TRIM25 expression correlated with a poor patient survival in HCC, indicating that oxidative stress was preferable for patients’ survival.

Of the HBV-producing proteins, HBx protein has been shown to be correlated with oxidative stress [50]. The mechanism involves HBx co-localization with mitochondria, and the C-terminal region has been shown to be critical for mitochondrial ROS production [51]. HBV with HBx protein expressed in mitochondria binds to voltage-dependent anion channels (VDAC3) and alters the mitochondrial transmembrane potential, resulting in ROS generation and the activation of several transcription factors [52]. Given that HBV may reside in the liver of billions of people worldwide, the changes in the oxidative stress status at different stages of the disease still need to be investigated.

#### 2.3.2. HCV-Related Chronic Hepatitis

##### Recent Advances in the Management of HCV-Related Chronic Hepatitis

Recently, it has been demonstrated that DAA treatment can achieve viral eradication in patients of highly aged patients or advanced cirrhosis patients [53]. After HCV eradication, the risk of HCC occurrence decreases [54]. However, as older patients and those with advanced disease can receive DAA treatment due to the low incidence of side effects, care for HCC is still very important [55].

Although long-term results are awaited, risk stratification for the risk of hepatocarcinogenesis and liver fibrosis after DAA treatment has been intensively investigated in many institutes. It is widely acknowledged that a historic interferon treatment-based sustained viral response (SVR) has reduced the risk of HCC; however, whether a DAA-induced SVR is the same or not is still in debate [56]. It has been reported that the risk of de novo HCC occurrence risk decreases by 50–78% in cirrhosis patients and 70–80% in non-cirrhosis patients. The absolute risk of HCC in patients with cirrhosis at the time of DAA treatment is reported to be 1.8–2.5%. The risk of HCC in non-cirrhosis patients is low, however, those with a high Fibrosis-4 (Fib-4) score, which represents cirrhotic status, have higher risk and require strict care [57,58]. A higher incidence of early occurrence of HCC after DAA treatment in patients with cirrhosis complicating undefined/non-malignant hepatic nodules has also been reported, indicating a new risk factor for HCC [59].

Several reports have assessed liver stiffness and the post-SVR clinical course in chronic hepatitis C. Liver stiffness, defined by TE, showed significant improvement after a DAA-induced SVR, especially in F4 stage cirrhosis patients, while it was less effective in patients with low platelet counts, indicating that severe fibrosis could not be improved [60]. From a report on the histological analysis of the resected liver, in patients who underwent liver resection due to HCC after an SVR, the liver stiffness measured by TE improved to the normal range, although histological fibrosis was evident as F4-cirrhosis [61]. A longitudinal analysis from Japan on liver stiffness before and after DAA showed that liver stiffness measurements decreased after an SVR24, while patients who developed HCC maintained higher liver stiffness than others [62]. In patients with higher liver stiffness, fibrosis-related activated hepatic stellate cells may be present; these cells produce cytokines, such as transforming growth factor (TGF)-β, which plays a prometastatic role in HCC by inducing epithelial-mesenchymal transition (EMT) [63]. An immunohistochemical study assessing fibrosis and LSEC capillarization before and after the administration of interferon/pegylated interferon induced-SVR showed that collagen (a fibrosis marker) decreased in 89% of patients, while, CD34 (an LSEC capillarization marker) did not change [64]. Residual capillarized LSECs after an SVR maybe a risk factor for hepatocarcinogenesis. Although there are no known effective markers of capillarized LSECs or thin fibrous septa, the platelet count and liver stiffness are candidate markers. Physicians should remember that a relatively high liver stiffness measurement is a risk factor for hepatocarcinogenesis and that F4 cirrhosis with fibrous septa might remain even after liver stiffness is reduced to the normal range (Figure 1).

In active cirrhosis with HCV-positive and active hepatitis, the LSECs were capillarized and the basement membrane was composed of activated HSCs and activated inflammatory T cells. After viral eradication with DAA therapy, patients with differentiated LSECs and quiescent HSCs have a lower risk of hepatocarcinogenesis, while those with capillarized LSECs and activated HSCs have a higher risk of hepatocarcinogenesis. HCV (hepatitis C virus), DAA (direct-acting antivirals), HSCs (hepatic stellate cells), LSECs (liver sinusoidal endothelial cells), CTL (cytotoxic T cells), Th (helper T cells)

##### Oxidative Stress in the Pathogenesis of HCV-Related Chronic Hepatitis and HCC

HCV itself has direct oncogenic potential, but the effect is weaker than that of HBV. HCV-related HCC involves the upregulation of many detoxification-related genes or immune response- related genes according to a microarray analysis, suggesting the strong impact of chronic inflammation and several stress responses in HCV-related hepatocarcinogenesis [65]. HCV antigens, especially core proteins, have been shown to be a major player in pathogenesis and hepatocarcinogenesis in infected hepatocytes in vitro [66]. Similar findings have been obtained in in vivo models. Indeed, an HCV core protein transgenic mouse model showed progressive hepatic steatosis resulting in HCC [67,68]. In HCV transgenic mice, the HCV core protein transgenic was shown to interact with mitochondria and oxidized the glutathione pool and reduced NADPH content. This indicated the direct role of oxidative stress induction [69]. HCV nonstructural (NS) proteins also have correlations with oxidative stress-related pathways in infected hepatocytes and HCC cells. One of the newly discovered host antioxidant pathways, the 3b-hydorxysterol-d24-reductase (DHCR24), has been shown to be induced via direct-interaction with HCV NS proteins, resulting in an antioxidant role via the scavenging of toxic ROS in HCV-infected cells [70]. This is to make HCV infected cells resistant against toxic ROS.

As oxidative stress can be produced via iron-induced Fenton reaction, iron overload has been shown to be an HCV-related disease-aggravating factor [71]. HCV proteins play a role in increasing hepatic iron levels by reducing the levels of hepcidine, a regulator of hepatic iron, thereby resulting in strong oxidative stress [72]. Additional administration of an iron-rich diet induced aggravation of hepatocarcinogenesis, suggesting the importance of dietary iron regulation to prevent hepatocarcinogenesis [73]. Given that HCV infection can induce hepatic steatosis and oxidative stress, chronic hepatitis C progression resembles NAFLD progression.

### 2.4. Oxidative Stress in the Pathogenesis of NAFLD-Related Steato-Hepatitis and HCC

NAFLD can be accompanied by chronic hepatitis B or C infection, resulting in an additional risk of chronic inflammation and HCC development. The mechanism underlying the progression of NAFLD is recognized as the “two-hits” theory: the first hit is lipid deposition, and the second hit involves several stress responses, including oxidative stress, ER stress, autophagy, and chronic inflammation related cytokines [74]. Given that there are several patterns of progression mechanisms aside from the two hits theory, “multiple parallel hits” theory has also been proposed [75]. These hits induce insulin resistance and lipid peroxidation, leading to liver steatosis and increased free fatty acid accumulation and resulting in the activation of the β-oxidation cycle in mitochondria [76]. Mitochondria play an important role in fat oxidation via the β-oxidation cycle and generate adenosine triphosphate (ATP).

The first hit of lipid deposition in the liver is induced via dietary intake. Advanced NAFLD patients have been shown to consume more carbohydrates/energy higher saturated fatty acid and lower polyunsaturated fatty acids [77,78]. Adipose tissue in NAFLD produces adipokines, such as tumor necrosis factor alpha (TNFα), resistin, and adiponectin. These factors induce chronic inflammation and insulin resistance. Overloaded free fatty acid induces mitochondria damage and generates ROS, which is the main source of oxidative stress.

Insulin resistance is characteristic of both type 2 diabetes and NAFLD progression. The liver is a background organ of systemic insulin resistance, which can develop via several mechanisms, such as liver-specific inflammation-related NF-κB activation followed by cyclic AMP (cAMP) signaling activation, hepatic glucose production and insulin substrate PDE3B inhibition [79], as well as liver-specific JNK activation followed by IRS-1 inhibition [80]. ROS are involved as inducers of such kinase activation in hepatocytes. Kupffer cells and macrophages in the liver are believed to play critical roles in insulin resistance. They are the initial drivers of hepatic inflammation, which become activated in the liver of obese patients. An immunohistochemical analysis of simple fatty liver and NASH revealed that the infiltration of CD68-positive macrophages, including Kupffer cells, was evident, even in the simple fatty liver stage of NAFLD [81]. Inactivating Kupffer cells and macrophages via M2 macrophage polarization can ameliorate obesity-induced insulin resistance and the endothelial ROS signaling pathway [82].

Hyperinsulinemia induces HCC development and progression via its hepatocyte proliferative function and favorable microenvironmental arrangement for tumorigenesis [83]. Recently, conserved AAA+ ATPases Pontin (RUVBL1) and Reptin (RUVBL2) have been shown to be associated with key cellular processes, such as chromatin remodeling, transcriptional regulation, mitosis, cell migration and insulin signaling [84,85]. They were both found to be up-regulated in HCC, and Pontin expression was shown to be a strong independent factor associated with a poor prognosis [86]. In a mouse model of liver-specific Pontin haploinsufficiency, Pontin was shown to regulate insulin signaling through the Akt/mTOR pathway [85]. The haploin sufficient model showed an initially delayed onset of HCC but later caught up, developing larger tumors. This controversial effect was explained by Pontin reduction and induced the worsening of insulin resistance and later HCC progression. Insulin resistance and its related pathways, such as the Akt/mTOR pathway, and the ATPases signaling correlated with the initiation and later progression of HCC should be clarified in detail.

The genetic characteristics of advanced NASH have been described. One of the first strong factors reported was the lipid storage-related patatin-like phospholipase domain-containing 3 (PNPLA3) [87]. A single nucleotide polymorphism (SNP) in this gene has been shown to be significantly correlated with NASH progression among races [88]. PNPLA3 is reportedly correlated with the liver fat concentration. Recently, several other genes have also been shown to be correlated with NAFLD progression. One of them, TM6SF2, has been shown to be correlated with the liver triglyceride content and very-low-density lipoprotein (VLDL) secretion [89]. Patients containing alleles for these disease-susceptible risk genes who have increased liver fat contents have shown lower plasma triglyceride levels and a preserved insulin sensitivity [90]. These patients are described as being “silent” on blood examinations, and their potential existence should be kept in mind when screening for NAFLD. These genetic analyses have indicated that a hepatic increase in triglyceride and free fatty acid levels can lead to advanced NASH, while oxidative stress-related genes are not genetically involved. This suggest that the increase in oxidative stress is secondary to the above genetic mechanisms, fatty liver, and insulin resistance.

NASH-HCC-related genes have also been suggested, although their influence is still not widely accepted. One of them, DYSF gene, which encodes dysferlin linked with skeletal muscle repair, has been found to have significantly increased levels in Japanese NAFLD patients [91]. Even hepatocarcinogenesis might be correlated with the ecological environment in NASH.

The oxidative stress status has been analyzed in NAFLD and NAFLD-related HCC. Limited antioxidant defenses contribute to the processes of both NASH and hepatocarcinogenesis [92,93]. In liver mitochondria from NASH patients and mouse models, ultrastructural alterations, impairment of ATP synthesis and increased production of ROS have been reported [94,95]. Iron facilitates the Fenton reaction, resulting in oxidative stress, just as with HCV pathogenesis. The effect of iron on NAFLD pathogenesis remains controversial. Levels of iron have shown to be elevated in NASH, and reducing iron levels has resulted in fair outcomes [96]. Iron absorption-related transporters in liver and intestine have been shown to be changed via high-cholesterol diet in a rat model [97]. Excess iron intake should be avoided in order to prevent NAFLD progression. However, one-third of early-stage NAFLD patients have been shown to be iron-deficient, and this condition was correlated with female gender, obesity and type 2 diabetes [98]. An on-demand approach regarding whether to reduce or increase iron intake is also required.

## 3. How to Manage Oxidative Stress in Pre-Cancer Stage

Given that oxidative stress is involved in chronic viral hepatitis and NAFLD cross-sectional pathology, the management strategy can be commonly applied. Nutritional support or supplementation is used to eliminate ROS or activate antioxidant pathways, such as the Nrf2 pathway [99]. In the case of the simultaneous activation of Nrf2 and NFkB, Nrf2 acts antagonistically against NFkB [100]. In a chemically induced liver fibrosis model, antioxidant pomegranate juice, which contained anthocyanins and hydrolysable tannins, reduced hepatic fibrosis via Nrf2 activation and NFkB inactivation [101]. Adjusting for the balance of inflammatory or oxidative stress responses and the antioxidant response in the presence of disease according to the patient’s oxidative stress condition is necessary; however, at present, this approach is not widely applied. The difficulty in defining the oxidative stress status is one reason for this. This background probably explains why the results of clinical studies of antioxidants to regulate carcinogenesis are often unsuccessful.

Trace elements are involved in oxidative stress-related conditions. Some of them, such as iron and zinc, are able to be measured as standard clinical markers and monitored. Iron has been shown to be toxic as an oxidative stress inducer in chronic liver disease, as mentioned in Section 2.4, and its level can be reduced by phlebotomy or iron chelator administration in chronic hepatitis and NASH [102]. Zinc plays a role in the reduction of inflammatory cytokines and oxidative stress via the synthesis of antioxidant enzymes and catalyzing enzymes, or by influencing transcriptional factors [103]. One of the inflammation related transcription factors NF-κB is reduced via its negative regulator zinc finger protein (A20) or PPAR-αα. An antioxidant enzyme Cu,Zn- SOD (SOD1) contains zinc as a co-factor. Zinc has been shown to be deficient in cases of chronic liver disease, especially cirrhosis, possibly because of the impaired absorption from the intestine and increased excretion in the urine [104]. Zinc is necessary for the function of Paneth cells, which prevent pathogenic microbial invasion in the intestine, a risk for subsequent hepatic inflammation via α-defensin production [105]. However, the presence of excess dietary zinc increases oxidative stress with an increased intestinal permeability that should be avoided [106]. An appropriate and effective supplementation strategy is therefore necessary, even for trace elements.

Selenium is one of the essential elements required for the normal development of human and animal organisms. Selenium activates GPx, which is a representative antioxidant enzyme. The Gpx-1 enzyme activity and mRNA levels decrease dramatically in a selenium deficient diet, whereas other selenoproteins are less sensitive [107]. Feeding a selenium deficient diet with glutathione deficiency resulted in oxidative stress, during which the protein malondialdehyde levels increase in the liver and an individual thus becomes sensitive to drug induced liver injury, thereby indicating the necessity of selenium for antioxidant system activation [108]. Given that the blood selenium level was observed to decrease in liver cirrhosis patients, supplementation may be one approach to improve the antioxidant function in such cases [109].

### 3.1. Dietary Intervention for Oxidative Stress

The potential dietary antioxidant intake has been assessed in several studies. Even an increase in the food frequency questionnaire-defined dietary total antioxidant capacity was shown to be correlated with a lower liver histological assessment of NASH-related hepatocellular ballooning [110]. The intake of orange juice, a source of flavonoids and vitamin C, for eight weeks, resulted in a reduction of total cholesterol, LDL-cholesterol, C reactive protein, and oxidative stress related markers, in a randomized study of 43 chronic hepatitis C patients [111]. Dietary vitamin C intake was shown to be inversely correlated with the presence of NAFLD, similarly to vitamin E, suggesting the favorable effect of both vitamins [112]. The positive effect of the intake of vitamin C on NAFLD prevention was shown to be dominant in middle-aged, non-obese males [113].

An iron-reduced diet, often coupled with phlebotomy, has been shown to be effective against chronic hepatitis C and NASH, resulting in a reduction of the risk of hepatocarcinogenesis [114]. Zinc supplementation has been evaluated in more than 1300 studies, although not many have shown statistically significant favorable results [115].

A small number of studies showed the preferable effects of zinc supplementation, suggesting the important role for antioxidant response. The transaminase level in chronic hepatitis C patients decreased [116], and the serum levels of type IV collagen and tissue inhibitors of matrix-metalloproteinase-1 (TIMP-1) levels in chronic hepatitis patients also decreased [117]. In cirrhosis patients, zinc supplementation may help to improve protein catabolism [118]. As zinc is often involved in standard laboratory examinations, to measure, evaluate, and adequate supplementation are thus all necessary steps. Large scale studies defining the best approach in chronic liver diseases are thus called for in the future.

Selenium supplementation has been shown to be effective in some patients with chronic thyroiditis, due to its immune targeting effect [119]. During chemotherapy for cancer patients, selenium supplementation has been shown to be associated with an improvement in fatigue, as well as in the liver and renal function [120]. However, in primary biliary cirrhosis, the supplementation of selenium did not show any antioxidant activities, while the renal excretion was increased, suggesting that a cirrhotic liver could not take advantage of selenium adequately [121]. Although selenium administration helps in the recovery of hepatic steatosis via PPAR-α activation in some diabetic mouse models [122], selenium supplementation to humans has been cautioned to increase the risk of type 2 diabetes [123]. It therefore remains difficult to draw any definite conclusions about selenium supplementation as an antioxidant.

### 3.2. Clinical Trials for Oxidative Stress

Many clinical trials have been undertaken to investigate whether antioxidants prevent cancer or death; however, the results are confusing. In the Alpha-Tocopherol, Beta-Carotene Cancer Prevention Study (ATBC), alpha-tocopherol reduced the incidence of prostate cancer, whereas beta-carotene increased the incidence of lung cancer and total mortality [124]. The Selenium and Vitamin E Cancer Prevention Trial (SELECT), a randomized control trial (RCT) that aimed to show the potential for vitamin E to reduce the risk of prostate cancer, showed a 17% increase in the incidence of prostate cancer [125]. Another study showed that beta-carotene supplementation was associated with an increased risk of lung cancer [126]. An epidemiologic study showed that dietary vitamin E intake and vitamin E supplement use was associated with a reduced risk of liver cancer, although vitamin C and multivitamin intake increased the risk of liver cancer [24]. To define the real effect of antioxidant supplementation, studies should be planned according to the oxidative stress-related conditions before the start of intervention.

Antioxidant therapy, such as the administration of vitamin E, has been shown to be effective in improving inflammation and histological activity in NASH and is recommended in several guidelines for NAFLD. However, the long-term effect of these therapies, including the beneficial effects on the risk of hepatocarcinogenesis, is unclear [127,128]. There are also other antioxidant agents that have been shown to have favorable effects on NASH and NASH-related hepatocarcinogenesis.

Antidiabetic agents are recommended for NAFLD patients complicated with diabetes. Most of the antidiabetic agents, but not insulin or insulin producers, have been shown to be effective for NAFLD. Metformin and pioglitazone have been accepted as representative antioxidant agents [129]. Metformin has also been shown to activate AMPK by inhibiting mitochondrial complex I and inducing AMPK-independent lysosomal changes, resulting in many favorable effects in carcinogenesis and the post-carcinogenesis control of cancers [130]. Metformin-related AMPK pathway activation is involved in many cell types, including T cells, B cells, hepatocytes, and even liver fibrosis-inducing hepatic stellate cells (HSCs). As an antioxidant agent, metformin activates the Nrf2 signaling pathway, resulting in the production of heme oxygenase-1 (HO-1; an antioxidant enzyme), in human endothelial cells and thereby increasing the antioxidant function of these cells. However, in several cancer cell lines, including HCC, metformin suppressed the Nrf2 expression in an AMPK-independent manner [131]. Research on the effect of metformin on normal and cancerous cells is still ongoing. At present, this agent is recognized as suitable for pre-cancer administration, to prevent hepatocarcinogenesis and post-cancer administration, to prevent HCC recurrence.

Peroxisome proliferator-activated receptors (PPARs) are nuclear receptors that play key roles in cellular metabolic homeostasis and inflammation. Pioglitazone is a diabetes agent that activates PPAR-γ. It has been shown to improve insulin resistance, and several studies reported a favorable effect on NASH. One six-month randomized study of pioglitazone plus hypocaloric diet showed a plasma aminotransferase decrease, insulin sensitivity improvement, hepatic fat content decrease, and a histopathological reduction in liver necro-inflammation [132]. However, an additional analysis of select patients in the same cohort revealed that pioglitazone induced whole-body weight gain, and the increased weight was due to an increase in adipose tissue mass and not water retention [133].

Other antioxidants have been shown to be effective in several small series of studies. The administration of L-carnitine (a mitochondrial long-chain fatty acid uptake-related molecule) was reported to be associated with the histological improvement of NASH in a mouse model [134] and an RCT [135]. Flavonoids (heterogeneous polyphenols) have been shown to exert an antioxidant function, protecting the liver in a CCl_4_-induced liver injury model [136]. A mixture of flavonolignans and minor polyphenolic compounds derived from the milk thistle plant (*Silybum marianum*) named silymarin has been shown to have antioxidant power [137]. The main component of silymarin, silybin, has been shown to restore nicotinamide adenine dinucleotide (NAD+) levels, decreasing the glucose uptake and lipid peroxidation and resulting in the improvement of NAFLD [138,139]. Silymarin was shown to be effective for improving NASH-related fibrosis in a randomized, double-blind, placebo-controlled study, although the number of patients was relatively small (49 for silymarin and 50 for placebo) [140].

Many studies using antioxidant agents have shown promising results for NAFLD, suggesting that these agents may be viable candidate compounds in addition to standard vitamin E.

## 4. How to Manage Oxidative Stress in the Cancer Stage

After the development of HCC, the role of oxidative stress changes. An adequate degree of oxidative stress should be maintained in order to regulate the progression of HCC. The ideal choice of agent for controlling chronic hepatitis or related hepatocarcinogenesis without suppressing the physiological roles of oxidative stress is difficult to determine.

### 4.1. Management of Oxidative Stress after Radical Therapy

After surgery or radio frequency ablation (RFA) treatment, HCC recurs in more than 60% of cases [141]. HCC recurrence is a critical factor associated with a poor survival of HCC. The recurrence pattern can be divided into two patterns: intrahepatic early metastasis and multicentric origin related late recurrence [142]. The phenotype of early-recurring tumors and the surrounding non-cancerous region has been described in several reports. An immunohistochemical analysis of tumors that recurred within 1 year after radical RFA showed that these tumors had a higher frequency (67%) of poorly differentiated type and higher rates of positivity for the proliferation marker Ki-67 than non-recurrence within 1 year, with a lower rate of positivity of the negative cell cycle regulator p27^Kip1^ [143]. A genome-wide gene-expression profile of resected cancer tissue and the surrounding noncancerous liver tissue showed that the increased expression of the cytochrome P450 1A2 (CYP1A2) gene in noncancerous tissue was a predictive marker for non-recurrence [144]. CYP1A2 is a form of hepatic cytochrome P450 oxidative system that is involved in drug and cholesterol metabolism. CYP1A2 knockout mice showed increased oxidative stress in liver microsomes, suggesting that CYP1A2 is an antioxidant molecule [145]. The plasma concentration of GPx3, another antioxidant enzyme, has been investigated in resected HCC [146]. Lower levels of plasma GPx3 were predictive of tumor progression and tumor recurrence. Based on these results of antioxidant-related marker analyses, the antioxidant reservoir function in the adjacent liver is critical for achieving a good survival after radical local treatment.

### 4.2. Management of Oxidative Stress in Combination with Anti-Cancer Agents

Many anti-cancer agents have been shown to induce oxidative stress to kill cancer cells. Recently, molecular-targeted agents, such as sorafenib, regorafenib and Lenvatinib, have become available as the standard treatment for HCC [147]. These agents mainly act on serine-threonine kinases, such as Raf-1, and on receptor tyrosine kinases, such as vascular endothelial growth factor receptor (VEGFR) and platelet-derived growth factor receptor-β (PDGFR-β), resulting in deficient tumor-related microvascular angiogenesis and proliferation of tumor cells. Oxidative stress induction for cancer cells has been shown to be one mechanism by which these agents exert their effects [148].

However, while these agents have shown promise for controlling advanced HCC, drug resistance is an increasing problem. A genome-wide CRISPR/Cas9-based screening of a sorafenib-treated HCC cell line identified KEAP1 as the top candidate drug resistance-related gene [149]. Given that KEAP1 disruption resulted in increased Nrf2 activity and Nrf2-driven genes and decreased ROS production, oxidative stress reduction is one of the mechanisms underlying resistance to sorafenib. Oxidative stress should therefore be maintained in order to ensure the ongoing efficacy of these agents.

Maintaining the antioxidant function in the area surrounding the tumor and maintaining oxidative stress in the tumor area are proposed strategy for controlling oxidative stress in HCC. It is not easy to foster such oxidative stress-related statuses through nutritional support. Eliminating oxidative stress, as is recommended in the pre-cancer stage, is conversely not recommended in the cancer stage (Figure 2).

In the pre-carcinogenesis stage (as chronic viral hepatitis and NAFLD), oxidative stress is usually increased via inflammation or obesity-related fat deposition. Oxidative stress is a toxic and disease-exacerbating factor that should be regulated in the pre-cancer stage. However, in the cancer stage and under treatment with molecular-target agents, such as sorafenib, regorafenib or Lenvatinib, oxidative stress is an important anti-cancer response and a pharmaceutically beneficial response. The antioxidant reservoir function in the adjacent liver is required to regulate recurrence and new carcinogenesis. This paradox is difficult to resolve.

## 5. Conclusions

Oxidative stress is a disease progression-related toxic response in cases of chronic liver diseases such as chronic hepatitis B and C and NASH. In chronic hepatitis B and C, the viral load and associated chronic inflammation and oxidative stress can now be considerably controlled; however, reduced (but still present) fibrosis and LSEC capillarization and associated HSC or Kupffer cell activation still lead to the development of HCC. In NASH, in particular, antioxidant agents, such as vitamin E, are widely accepted as effective for controlling disease progression. However, after cancer development, oxidative stress is an anti-cancer response regulating cancer cell growth or proliferation and works as an adjuvant agent for anti-cancer molecular-targeted agents. Defining appropriate clinical characteristics (e.g., low platelet counts or high liver stiffness measurements before viral eradication and oxidative stress-related stages) before and after hepatocarcinogenesis will be necessary to improve the prognosis of patients with chronic liver disease.

## Figures and Tables

**Figure 1 nutrients-12-01576-f001:**
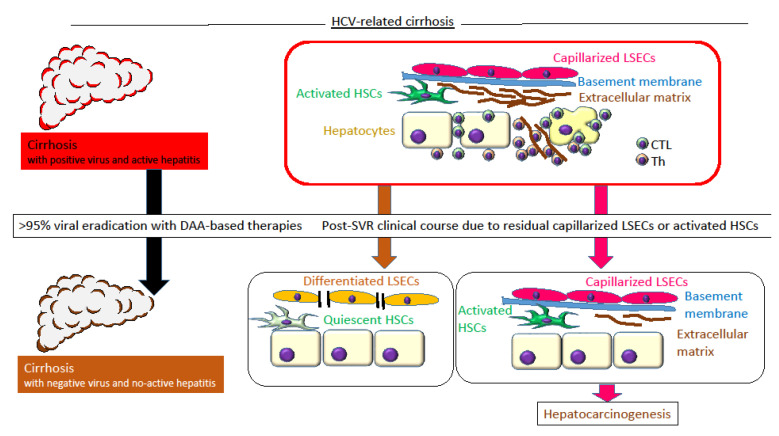
A conceptual diagram of probable post-sustained virologic response (SVR) hepatocarcinogenesis in hepatitis C virus (HCV)-related cirrhosis.

**Figure 2 nutrients-12-01576-f002:**
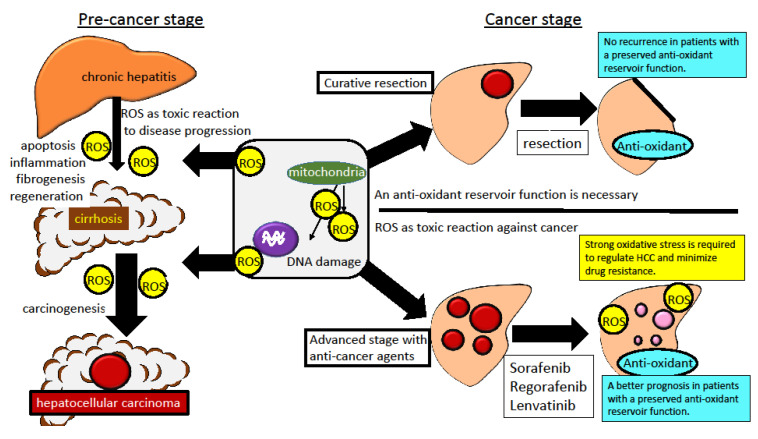
A conceptual diagram of oxidative stress in the pre- and post-carcinogenesis stage.

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
