# Peer review of "Oxidative Stress Management in Chronic Liver Diseases and Hepatocellular Carcinoma"

_nutrients, 2020, doi:10.3390/nu12061576_

Round 1

Reviewer 1 Report

The present review article entitled “Oxidative stress management in chronic liver diseases and hepatocellular carcinoma” presents with the important and inspiring updates on the counterbalancing roles of oxidative stress in the pathogenesis of HBV or HCV-related chronic hepatitis and hepatocellular carcinoma, and on the management in precancerous and cancer stages.

# Major concerns and recommendations:

1.In my (our) eyes, it is uncomfortable with the antiviral treatments-oxidative stress-bidirectional fibrogenesis-carcinogenesis axis missed when tackling chronic hepatitis B and C.

2.The cell localization and specification for the addressed pathways are not fully clear throughout the review article.

The authors’ statements somewhat applied to NAFLD alone. Currently, it is well-known that novel antiviral treatments can eradicate or ameliorate CHC and CHB worldwide. Concise yet comprehensive, solid and impressive updates relevant to impacts of antiviral treatments/liver fibrosis regression/carcinogenesis risk reduction… are highly expected among us gastroenterologist readers and researchers dealing with the treatments/kinetics in hepatic necroinflammatory resolution/fibrosis regression/stiffness declines/posttreatment histology- and stiffness-based fibrosis staging/liver histology and marker quantitation/post viral-eradication surveillance/pretreatment and posttreatment event risk estimation modeling… along the treatment timeline. As you can see, constantly we cannot spare the “longitudinal kinetics” with “antiviral treatments” across the time frames indexed from the antiviral treatment baseline in daily practice. It can be performed with ease for you to concisely tackle the in-depth concepts regarding the impacts of viral eradication. The present review article needs to be extensively rewritten incorporating the aforementioned (impacts of antiviral treatments/clear temporal and causal relationships/fibrogenesis…) into every section tackling CHB or CHC in addition to the important updates on oxidative stress. Besides, remind the readers of the cell identification (parenchymal hepatocytes? hepatoma cells? quiescent hepatic stellate cells?...) and localizations from time to time wherever appropriate. Thank you.    

References, for example:

- Chen SH, Lai HC, Chiang IP, et al. Performance of acoustic radiation force impulse elastography for staging liver fibrosis in patients with chronic hepatitis C after viral eradication. Clin Infect Dis. 2020;70:114–122.

- Jacobson IM, Lim JK, Fried MW. American Gastroenterological Association Institute Clinical Practice Update- expert review: Care of patients who have achieved a sustained virologic response after antiviral therapy for chronic hepatitis C infection. Gastroenterology. 2017;152:1578–1587.

- Martínez-Campreciós J, Bonis Puig S, Pons Delgado M, Salcedo Allende MT, Mínguez Rosique B, Genescà Ferrer J. Transient elastography in DAA era. Relation between post-SVR LSM and histology. J Viral Hepat. 2020;27(4):453–455. doi:10.1111/jvh.13245

- Knop V, Mauss S, Goeser T, et al. Dynamics of liver stiffness by transient elastography in patients with chronic hepatitis C virus infection receiving direct-acting antiviral therapy-Results from the German Hepatitis C-Registry.  J Viral Hepat. 2020; [published online ahead of print]

- Ogasawara N, Saitoh S, Akuta N, et al. Advantage of liver stiffness measurement before and after direct-acting antiviral therapy to predict hepatocellular carcinoma and exacerbation of esophageal varices in chronic hepatitis C. Hepatol Res. 2019;[published online ahead of print].

- Elshaarawy O, Mueller J, Guha IN, et al. Spleen stiffness to liver stiffness ratio significantly differs between ALD and HCV and predicts disease-specific complications. JHEP Rep. 2019;1:99–106.

- Nishio T, Hu R, Koyama Y, Liang S, Rosenthal SB, Yamamoto G, et al. Activated hepatic stellate cells and portal fibroblasts contribute to cholestatic liver fibrosis in MDR2 knockout mice. J Hepatol. 2019; 71:573–585.

- Poisson J, Lemoinne S, Boulanger C, Durand F, Moreau R, Valla D, et al. Liver sinusoidal endothelial cells: Physiology and role in liver diseases. J Hepatol. 2017; 66:212–227.

- Tsuji N, Ishiguro S, Sasaki Y, Kudo M. CD34 expression in noncancerous liver tissue predicts multicentric recurrence of hepatocellular carcinoma. Dig Dis. 2013; 31:467–471.

- Luangmonkong T, Suriguga S, Mutsaers HAM, Groothuis GMM, Olinga P, Boersema M. Targeting Oxidative Stress for the Treatment of Liver Fibrosis. Rev Physiol Biochem Pharmacol. 2018;175:71–102.

- Rebbani K, Tsukiyama-Kohara K. HCV-Induced Oxidative Stress: Battlefield-Winning Strategy. Oxid Med Cell Longev. 2016;2016:7425628.

# Minor concerns and recommendations:

1.Replace most of the old references, including No.1, 2, 4, 6, 7…81, with the latest representative ones if any. Please try citing the newest references throughout the entire article. Citing old references does not justify future studies are warranted for the currently unknown (e.g. Lines 129–131).

2.Replace Figure 1 with another, conveying comprehensive, high-quality and impressive impacts as you previously published. The current redundant figure does not fit into the current relevance and critical reviews.

3.Incorporate fibrogenesis into Figure 2 as appropriate and capable.

4.Should Line 185 read-- NASH-HCC related genes “have”?

5.Line 219 supplementation might be useful only for “select”? populations, carrying a general risk for cancer

6.Lines 220, 221 Clarifying these conflicting indications for oxidative stress management is necessary but very difficult/ Line 316 This paradox is difficult to resolve – please address concise, solid, substantial and formal explanations and recommendations as capable.

7.Line 263 Many studies using antioxidant agents have shown “promising” results for NAFLD 

8.Lines 323, 324, 325 Defining appropriate oxidative stress-related stages of chronic liver diseases, including HCC, will be necessary to improve the prognosis of chronic liver disease-- What stages (are you referring to the pre-cancer and cancer stages in Figure 2 alone?)? How about incorporating (discussing) the staging from the pretreatment baseline to post viral eradication status? What will the impacts (on oxidative stress, fibrogenesis…) of antiviral treatments be like across the antiviral treatment timeline (key issues we care about in daily practice and research settings)? Related introduction, reviews or citations are not seen in the present article?  

  1. A ”Conclusion” section is needed?

Author Response

Comments and Suggestions for Authors 1

Response

The present review article entitled “Oxidative stress management in chronic liver diseases and hepatocellular carcinoma” presents with the important and inspiring updates on the counterbalancing roles of oxidative stress in the pathogenesis of HBV or HCV-related chronic hepatitis and hepatocellular carcinoma, and on the management in precancerous and cancer stages.

# Major concerns and recommendations:

  1. In my (our) eyes, it is uncomfortable with the antiviral treatments-oxidative stress-bidirectional fibrogenesis-carcinogenesis axis missed when tackling chronic hepatitis B and C.

Response: We added the following new sections to explain the recent advances in the treatment and residual problems of hepatocarcinogenesis and its mechanisms, leading to an oxidative stress management strategy for these patients: 2.3.1.1 Recent advances in the management of HBV related chronic hepatitis and 2.3.2.1. Recent advances in the management of HCV related chronic hepatitis.

2.The cell localization and specification for the addressed pathways are not fully clear throughout the review article.

Response: We added the following new section to explain the liver parenchymal and non-parenchymal cells mechanisms on chronic hepatitis progression and hepatocarcinogenesis. recent advances in the treatment: 2.1. Oxidative stress in the cellular pathogenesis of chronic liver diseases.

The authors’ statements somewhat applied to NAFLD alone. Currently, it is well-known that novel antiviral treatments can eradicate or ameliorate CHC and CHB worldwide. Concise yet comprehensive, solid and impressive updates relevant to impacts of antiviral treatments/liver fibrosis regression/carcinogenesis risk reduction… are highly expected among us gastroenterologist readers and researchers dealing with the treatments/kinetics in hepatic necroinflammatory resolution/fibrosis regression/stiffness declines/posttreatment histology- and stiffness-based fibrosis staging/liver histology and marker quantitation/post viral-eradication surveillance/pretreatment and posttreatment event risk estimation modeling… along the treatment timeline. As you can see, constantly we cannot spare the “longitudinal kinetics” with “antiviral treatments” across the time frames indexed from the antiviral treatment baseline in daily practice. It can be performed with ease for you to concisely tackle the in-depth concepts regarding the impacts of viral eradication. The present review article needs to be extensively rewritten incorporating the aforementioned (impacts of antiviral treatments/clear temporal and causal relationships/fibrogenesis…) into every section tackling CHB or CHC in addition to the important updates on oxidative stress. Besides, remind the readers of the cell identification (parenchymal hepatocytes? hepatoma cells? quiescent hepatic stellate cells?...) and localizations from time to time wherever appropriate. Thank you.  

Response: We added the following new sections to explain the recent advances in the treatment and residual problems of hepatocarcinogenesis and its mechanisms leading to oxidative stress management strategy for these patients: 2.3.1.1 Recent advances in the management of HBV related chronic hepatitis and 2.3.2.1. Recent advances in the management of HCV related chronic hepatitis. We added the following new section to explain the mechanisms through which liver parenchymal and non-parenchymal cells are involved in the progression of chronic hepatitis and hepatocarcinogenesis, as well as recent advances in the treatment: 2.1. Oxidative stress in the cellular pathogenesis of chronic liver diseases.

References, for example:

- Chen SH, Lai HC, Chiang IP, et al. Performance of acoustic radiation force impulse elastography for staging liver fibrosis in patients with chronic hepatitis C after viral eradication. Clin Infect Dis. 2020;70:114–122.

- Jacobson IM, Lim JK, Fried MW. American Gastroenterological Association Institute Clinical Practice Update- expert review: Care of patients who have achieved a sustained virologic response after antiviral therapy for chronic hepatitis C infection. Gastroenterology. 2017;152:1578–1587. 

- Martínez-Campreciós J, Bonis Puig S, Pons Delgado M, Salcedo Allende MT, Mínguez Rosique B, Genescà Ferrer J. Transient elastography in DAA era. Relation between post-SVR LSM and histology. J Viral Hepat. 2020;27(4):453–455. doi:10.1111/jvh.13245

- Knop V, Mauss S, Goeser T, et al. Dynamics of liver stiffness by transient elastography in patients with chronic hepatitis C virus infection receiving direct-acting antiviral therapy-Results from the German Hepatitis C-Registry. J Viral Hepat. 2020; [published online ahead of print]

- Ogasawara N, Saitoh S, Akuta N, et al. Advantage of liver stiffness measurement before and after direct-acting antiviral therapy to predict hepatocellular carcinoma and exacerbation of esophageal varices in chronic hepatitis C. Hepatol Res. 2019;[published online ahead of print].

- Elshaarawy O, Mueller J, Guha IN, et al. Spleen stiffness to liver stiffness ratio significantly differs between ALD and HCV and predicts disease-specific complications. JHEP Rep. 2019;1:99–106.

- Nishio T, Hu R, Koyama Y, Liang S, Rosenthal SB, Yamamoto G, et al. Activated hepatic stellate cells and portal fibroblasts contribute to cholestatic liver fibrosis in MDR2 knockout mice. J Hepatol. 2019; 71:573–585.

- Poisson J, Lemoinne S, Boulanger C, Durand F, Moreau R, Valla D, et al. Liver sinusoidal endothelial cells: Physiology and role in liver diseases. J Hepatol. 2017; 66:212–227.

- Tsuji N, Ishiguro S, Sasaki Y, Kudo M. CD34 expression in noncancerous liver tissue predicts multicentric recurrence of hepatocellular carcinoma. Dig Dis. 2013; 31:467–471.

- Luangmonkong T, Suriguga S, Mutsaers HAM, Groothuis GMM, Olinga P, Boersema M. Targeting Oxidative Stress for the Treatment of Liver Fibrosis. Rev Physiol Biochem Pharmacol. 2018;175:71–102.

- Rebbani K, Tsukiyama-Kohara K. HCV-Induced Oxidative Stress: Battlefield-Winning Strategy. Oxid Med Cell Longev. 2016;2016:7425628.

# Minor concerns and recommendations:

1.Replace most of the old references, including No.1, 2, 4, 6, 7…81, with the latest representative ones if any. Please try citing the newest references throughout the entire article. Citing old references does not justify future studies are warranted for the currently unknown (e.g. Lines 129–131).

Response: We added the above recommended references and additional new references as recommended.

  1. Replace Figure 1 with another, conveying comprehensive, high-quality and impressive impacts as you previously published. The current redundant figure does not fit into the current relevance and critical reviews.

Response: We have replaced Figure 1 with another figure explaining post-SVR hepatocarcinogenesis related to LSEC and HSC activation.

3.Incorporate fibrogenesis into Figure 2 as appropriate and capable.

Response: We added “fibrogenesis” in Figure 2 as indicated.

4.Should Line 185 read-- NASH-HCC related genes “have”?

Response: We changed “has” to “have” as you indicated.

5.Line 219 supplementation might be useful only for “select”? populations, carrying a general risk for cancer

Response: We changed this sentence as follows:

To define the real effect of antioxidant supplementation, studies should be planned according to the oxidative stress-related conditions before the start of intervention.

6.Lines 220, 221 Clarifying these conflicting indications for oxidative stress management is necessary but very difficult/ Line 316 This paradox is difficult to resolve – please address concise, solid, substantial and formal explanations and recommendations as capable.

Response:

Line 220,221 We changed this sentence as follows:

To define the real effect of antioxidant supplementation, studies should be planned according to the oxidative stress-related conditions before the start of intervention.

Line 316 We changed this sentence as follow.

7.Line 263 Many studies using antioxidant agents have shown “promising” results for NAFLD

Response: We changed “good” to “promising” as indicated.

8.Lines 323, 324, 325 Defining appropriate oxidative stress-related stages of chronic liver diseases, including HCC, will be necessary to improve the prognosis of chronic liver disease-   - What stages (are you referring to the pre-cancer and cancer stages in Figure 2 alone?)? How     about incorporating (discussing) the staging from the pretreatment baseline to post viral eradication status? What will the impacts (on oxidative stress, fibrogenesis…) of antiviral treatments be like across the antiviral treatment timeline (key issues we care about in daily practice and research settings)? Related introduction, reviews or citations are not seen in the present article?

Response:

We changed the conclusion section to also explain on the the risk stratification and the management strategy for oxidative stress in chronic hepatitis B and C (Lines 433-443)

  1. A ”Conclusion” section is needed?

Response: We changed the “Discussion” section to “Conclusion” as we think this section is conclusively discussed about the contents.

Reviewer 2 Report

In this review, Uchida et al., and colleagues have highlighted the differential effects of oxidative stress in chronic liver diseases pre- and post-carcinogenesis. The authors also discuss the involvement of oxidative stress in the pathogenesis of NAFLD and steatohepatitis. Finally, they discussed the strategies, how to manage oxidative stress in pre-cancer and cancer stage. Overall, the review is nicely written providing sufficient references and good graphical representation of the ideas to be conveyed.

I only have some minor suggestions to be addressed and incorporated in this review.

  1. Include some aspects of NF-κB activation by oxidative stress. NF-κB activation by free radicals can cause various biological effects. For example, NF-κB activation by superoxide dismutase (SOD2) has been associated with lung adenocarcinoma progression and poor prognosis [PMID: 23784082]. It’s involvement in hepatocellular carcinoma /NAFLD /hepatitis can be discussed. Furthermore, oxidative stress generated in tumors can impair NF-κB translocation in different cell types.
  2. Vitamins (ex., B6, C, E) are known as major natural antioxidants. Discussing their effects on managing oxidative stress would definitely add more to our understanding of their role in these chronic pathologies.

Suggested readings for NF-kB and oxidative stress, PMID: 28060562; PMID: 8891667; PMID: 9706220

Author Response

Comments and Suggestions for Authors 2

Response

In this review, Uchida et al., and colleagues have highlighted the differential effects of oxidative stress in chronic liver diseases pre- and post-carcinogenesis. The authors also discuss the involvement of oxidative stress in the pathogenesis of NAFLD and steatohepatitis. Finally, they discussed the strategies, how to manage oxidative stress in pre-cancer and cancer stage. Overall, the review is nicely written providing sufficient references and good graphical representation of the ideas to be conveyed.

I only have some minor suggestions to be addressed and incorporated in this review.

  1. Include some aspects of NF-κB activation by oxidative stress. NF-κB activation by free radicals can cause various biological effects. For example, NF-κB activation by superoxide dismutase (SOD2) has been associated with lung adenocarcinoma progression and poor prognosis [PMID: 23784082]. It’s involvement in hepatocellular carcinoma /NAFLD /hepatitis can be discussed. Furthermore, oxidative stress generated in tumors can impair NF-κB translocation in different cell types.

Response:

We added an explanation on the effect of NF-kB on Kupffer cell activation in the 2.1. Oxidative stress in the cellular pathogenesis of chronic liver diseases section (Lines 304-307), and in the 3. How to manage oxidative stress in the pre-cancer stage section (Lines 304-307).

  1. Vitamins (ex., B6, C, E) are known as major natural antioxidants. Discussing their effects on managing oxidative stress would definitely add more to our understanding of their role in these chronic pathologies.

Response:

We added an explanation about vitamins to a new additional section: 3.1. Dietary intervention for oxidative stress.

Suggested readings for NF-kB and oxidative stress, PMID: 28060562; PMID: 8891667; PMID: 9706220

Round 2

Reviewer 1 Report

-Figure 1: heading: HCV-related cirrhosis

-Figure 1: Many mechanisms accounting for the post-SVR carcinogenesis are still under study. The heading would read: A conceptual diagram of probable post-SVR hepatocarcinogenesis in HCV-related cirrhosis

-Figure 1: in previously published reports, DAA-based therapies, “post-SVR”, “D”ifferentiated LSECs, “Q”uiescent HSCs, LSEC”s”, HSC”s”, “H”epatocarcinogenesis… are commonly seen (judging from Shetty S, Lalor PF, Adams DH. Liver sinusoidal endothelial cells - gatekeepers of hepatic immunity. Nat Rev Gastroenterol Hepatol. 2018;15:555–567. Sun LJ, Yu JW, Shi YG, Zhang XY, Shu MN, Chen MY. Hepatitis C virus core protein induces dysfunction of liver sinusoidal endothelial cell by down-regulation of silent information regulator 1. J Med Virol. 2018;90:926–935. Hayes CN, Zhang P, Zhang Y, Chayama K. Molecular Mechanisms of Hepatocarcinogenesis Following Sustained Virological Response in Patients with Chronic Hepatitis C Virus Infection. Viruses. 2018;10:531.)

-Figure 1: not “collagen fiber” alone. Replace “Collagen fiber” with “Extracellular matrix”? (Researchers work on other ECM and BM components, e.g. elastins, laminin… too)

-Figure 2: a typographical error “[”?

-Line 228: “an SVR to interferon-/ pegylated interferon-based therapies” was addressed in reference No 58 published in 2012 (not DAA)

Author Response

Comments and Suggestions for Authors 1 (second revise)

Comment

-Figure 1: heading: HCV-related cirrhosis

Response: We added the heading “HCV-related cirrhosis” as suggested.

Comment

-Figure 1: Many mechanisms accounting for the post-SVR carcinogenesis are still under study. The heading would read: A conceptual diagram of probable post-SVR hepatocarcinogenesis in HCV-related cirrhosis

Response: We changed the heading of the Figure 1 legend as suggested.

Comment

-Figure 1: in previously published reports, DAA-based therapies, “post-SVR”, “D”ifferentiated LSECs, “Q”uiescent HSCs, LSEC”s”, HSC”s”, “H”epatocarcinogenesis… are commonly seen (judging from Shetty S, Lalor PF, Adams DH. Liver sinusoidal endothelial cells - gatekeepers of hepatic immunity. Nat Rev Gastroenterol Hepatol. 2018;15:555–567. Sun LJ, Yu JW, Shi YG, Zhang XY, Shu MN, Chen MY. Hepatitis C virus core protein induces dysfunction of liver sinusoidal endothelial cell by down-regulation of silent information regulator 1. J Med Virol. 2018;90:926–935. Hayes CN, Zhang P, Zhang Y, Chayama K. Molecular Mechanisms of Hepatocarcinogenesis Following Sustained Virological Response in Patients with Chronic Hepatitis C Virus Infection. Viruses. 2018;10:531.)

Response: We changed these terms as suggested.

Comment

-Figure 1: not “collagen fiber” alone. Replace “Collagen fiber” with “Extracellular matrix”? (Researchers work on other ECM and BM components, e.g. elastins, laminin… too)

Response: We changed “Collagen fiber” to “Extracellular matrix” as suggested.

Comment

-Figure 2: a typographical error “[”?

Response: Yes, it was a typing error. We removed that.

Comment

-Line 228: “an SVR to interferon-/ pegylated interferon-based therapies” was addressed in reference No 58 published in 2012 (not DAA)

Response: Yes, that was our mistake. We changed the sentence as indicated (line 227-230).